# The Effects of Mitochondrial Transplantation on Sepsis Depend on the Type of Cell from Which They Are Isolated

**DOI:** 10.3390/ijms241210113

**Published:** 2023-06-14

**Authors:** Yun-Seok Kim, Han A Reum Lee, Min Ji Lee, Ye Jin Park, Sehwan Mun, Chang June Yune, Tae Nyoung Chung, Jinkun Bae, Mi Jin Kim, Yong-Soo Choi, Kyuseok Kim

**Affiliations:** 1Department of Emergency Medicine, CHA University School of Medicine, Seongnam 13497, Gyeonggi, Republic of Korea; loupys@naver.com (Y.-S.K.); harlee91@naver.com (H.A.R.L.); minji.lee29@gmail.com (M.J.L.); yejin6577@naver.com (Y.J.P.); bogmi0415@gmail.com (S.M.); june1976@hanmail.net (C.J.Y.); hendrix74@cha.ac.kr (T.N.C.); galen97@chamc.co.kr (J.B.); 2Department of Emergency Medicine, CHA Bundang Medical Center, CHA University, Seongnam 13497, Gyeonggi, Republic of Korea; 3Department of Biotechnology, CHA University, Seongnam 13497, Gyeonggi, Republic of Korea; treasure7744@naver.com (M.J.K.); yschoi6088@gmail.com (Y.-S.C.)

**Keywords:** sepsis, immune modulation, hyperinflammation, immune paralysis, mitochondria transplantation, mitochondria dysfunction

## Abstract

Previously, we have shown that mitochondrial transplantation in the sepsis model has immune modulatory effects. The mitochondrial function could have different characteristics dependent on cell types. Here, we investigated whether the effects of mitochondrial transplantation on the sepsis model could be different depending on the cell type, from which mitochondria were isolated. We isolated mitochondria from L6 muscle cells, clone 9 liver cells and mesenchymal stem cells (MSC). We tested the effects of mitochondrial transplantation using in vitro and in vivo sepsis models. We used the LPS stimulation of THP-1 cell, a monocyte cell line, as an in vitro model. First, we observed changes in mitochondrial function in the mitochondria-transplanted cells. Second, we compared the anti-inflammatory effects of mitochondrial transplantation. Third, we investigated the immune-enhancing effects using the endotoxin tolerance model. In the in vivo polymicrobial fecal slurry sepsis model, we examined the survival and biochemical effects of each type of mitochondrial transplantation. In the in vitro LPS model, mitochondrial transplantation with each cell type improved mitochondrial function, as measured by oxygen consumption. Among the three cell types, L6-mitochondrial transplantation significantly enhanced mitochondrial function. Mitochondrial transplantation with each cell type reduced hyper-inflammation in the acute phase of in vitro LPS model. It also enhanced immune function during the late immune suppression phase, as shown by endotoxin tolerance. These functions were not significantly different between the three cell types of origin for mitochondrial transplantation. However, only L6-mitochondrial transplantation significantly improved survival compared to the control in the polymicrobial intraabdominal sepsis model. The effects of mitochondria transplantation on both in vitro and in vivo sepsis models differed depending on the cell types of origin for mitochondria. L6-mitochondrial transplantation might be more beneficial in the sepsis model.

## 1. Introduction

Mitochondria are key components in cellular metabolism, cell growth, apoptosis, calcium homeostasis, redox status, etc., and their dysfunction has been implicated as a therapeutic target for various diseases [1]. Mitochondrial transplantation has been first proposed to be useful for the treatment of ischemia-reperfusion injury of the heart, and it shows promising clinical benefits regarding neonatal congenital heart diseases [2,3].

Mitochondrial transplantation enhances oxygen consumption, ATP synthesis, cell viability and post-infarct cardiac function [3]. Given these interesting findings, many diseases are being investigated as candidates for mitochondrial transplantation in both preclinical and clinical studies [4]. In sepsis, mitochondrial damage is critical, and therapeutic drugs might be used to enhance mitochondrial function. However, in cases of irreversible damage, mitochondrial transplantation could be a novel strategy [5]. We previously showed that mitochondrial transplantation could have immune modulation effects in the sepsis model [6]. Mitochondria are known to be different in terms of number, size, shape, distribution, and feature from cell to cell [7]. Therefore, the identification of the best mitochondrial donor cell is a very important issue in mitochondrial transplantation [8]. Skeletal muscle and MSCs have been suggested as an appropriate candidate for mitochondrial isolation [7]. Skeletal muscle has plenty of mitochondria, so it could be a good source cell for mitochondrial transplantation. MSCs are present in a quiescent state and appear to be primarily glycolytic. However, when transferred to the nutrient-rich artificial culture environment, MSCs become dependent on oxidative phosphorylation [9]. MSC has advantages for cell therapy or mitochondrial transplantation: viability and regenerative capacity after preservation at −80 °C; simplicity of isolation and cryopreservation; rapid replication; and minimal immunoreactivity [10]. Hepatocytes are enriched with mitochondria that comprise 13–20% of the liver volume [11].

With this in mind, we hypothesized that the effects of transplanting mitochondria would vary depending on the cell type from which they were isolated. To test this hypothesis, we used both in vivo and in vitro sepsis models. We isolated mitochondria from three different cell lines: L6 muscle cell line, clone 9 hepatocyte cell line, and MSC, mesenchymal stem cell. We examined the immunomodulatory effects and survival of mitochondrial transplantation using different cell types for mitochondrial isolation (Figure 1).

## 2. Results

### 2.1. ATP Content and Synthesis

Mitochondria isolated from three types of cells (L6, C9, MSC) had a protein mass of 10 µg and ATP content and synthesis were measured, as shown in Figure 2. In terms of ATP content, the values for L6 and C9 were similar and measured lower than that of MSC. However, in the ATP synthesis, MSC exhibited the lowest measurement, while L6 and C9 showed higher measurements. Therefore, we confirmed that L6 and C9 had a lower ATP content but a higher capacity for ATP synthesis compared to MSCs (Figure 2).

### 2.2. The Effect of Mitochondrial Transplantation on Mitochondrial Functions: In Vitro LPS Stimulation of THP-1

When we measured Basal OCR, ATP-linked respiration, and maximal respiration with a Seahorse XF analyzer and a Seahorse XF Cell Mito Stress Test Kit (Agilent, Santa Clara, CA, USA), we found that mitochondrial transplantation significantly increased mitochondrial respiration in the in vitro LPS stimulation of THP-1. Mitochondrial function was increased with L6 mitochondrial transplantation (Figure 3A,B).

### 2.3. The Immune Modulation Effects of Mitochondrial Transplantation: In Vitro LPS Stimulation of THP-1

To investigate the effects of mitochondrial transplantation on immune modulation in THP-1 cells, we stimulated THP-1 cells using LPS. LPS was used to induce hyper-inflammation and endotoxin tolerance conditions. Subsequently, 10 µg of mitochondria obtained from each cell type was transplanted into THP-1 cells, and the level of TNF-alpha, an inflammatory cytokine, was measured using ELISA (ab181421, Abcam, Waltham, MA, USA). During the hyperinflammatory phase, mitochondrial transplantation significantly decreased TNF-alpha levels. However, under endotoxin tolerance, it led to an increase in TNF-alpha levels (Figure 4).

### 2.4. Survival Study According to the Injection of Mitochondria on Sepsis

When we evaluated the effects of mitochondrial transplantation on the survival of a polymicrobial sepsis model, only L6 mitochondrial transplantation demonstrated a significant increase in survival compared to the control (Figure 5A).

### 2.5. Plasma Chemistry Assay: ALT and Creatinine

Plasma creatinine and ALT were measured in the in vivo sepsis model following L6-mitochondrial transplantation. They indicated the favorable effects of mitochondrial transplantation in the sepsis model (Figure 5B,C)

### 2.6. Arterial Blood Gas Analysis

There were no significant differences in the results of arterial blood gas analyses between the sepsis group and sepsis with the mitochondrial transplantation group (Appendix A).

## 3. Discussion

This study is the first to demonstrate the effects of mitochondrial transplantation on sepsis models could vary depending on the cell type, from which mitochondria were isolated. Additionally, we showed that the oxidative phosphorylation function (oxygen consumption rate) of mitochondria differs depending on their origin cell. L6-mitochondria exhibited superior mitochondria function when transplanted into in vitro sepsis model in terms of oxidative phosphorylation. Furthermore, L6-mitochondrial transplantation displayed a more pronounced survival effect in the in vivo sepsis model, compared to MSC or clone 9 cell.

Mitochondria are found in all mammalian cells except in red blood cells [12]. Classically, the function of mitochondria was considered to produce energy factories. However, it has been known that mitochondria have multiple important functions such as calcium homeostasis, redox status, control of apoptosis, cell proliferation, or immune modulation effect [13]. Mitochondrial dysfunction was considered the main pathophysiology in various important diseases [1,14]. With this background, mitochondrial transplantation has been widely investigated as a therapeutic strategy for many mitochondrial dysfunction [12,15]. To perform mitochondria transplantation, it is necessary to isolate healthy mitochondria from cells, which have different characteristics in terms of number, size, shape, distribution, and functions [7]. Consequently, the effects of mitochondria transplantation might be different depending on the specific cell types from which the mitochondria are isolated.

Previously, we demonstrated the immune modulation effects of mitochondria transplantation on both in vivo and in vitro sepsis models using L6 cells for mitochondria isolation. In the present study, we utilized different healthy cells to isolate the mitochondria, and observed that L6-mitochondria exhibited superior effects when transplanted on the in vitro sepsis model in terms of mitochondrial oxidative phosphorylation. These findings were consistent in the fecal slurry sepsis animal model, with a higher survival rate.

The effects of mitochondrial transplantation were investigated both in terms of oxidative phosphorylation and immune modulation. We hypothesized that mitochondrial transplantation would enhance mitochondrial oxidative phosphorylation, which in turn would promote immune modulation. However, contrary to our expectations, these effects were not correlated. L6-mitochondrial transplantation exhibited a significantly greater enhancement of oxidative phosphorylation compared to clone 9- or MSC-mitochondrial transplantation, but there were no significant differences in immune modulation effects. Interestingly, L6-mitochondrial transplantation resulted in a higher survival rate compared to clone 9- or MSC-mitochondrial transplantation. We did not investigate the exact mechanism by which mitochondrial transplantation influences oxidative phosphorylation. It might simply supply oxidative phosphorylation enzymes although other mechanisms may also be involved. Further investigation is needed to explore these possibilities.

Mitochondria transplantation has been studied in various diseases, especially ischemia-reperfusion injury [7,16]. Currently, it is unknown whether the transplantation of L6 mitochondira would have better effects on ischemia-reperfusion injury. Further experiments are needed to discover this.

This study has several limitations. First, we only tested three cell types for mitochondrial isolation. More cell types are needed to determine the best candidate cells for mitochondria isolation. Second, the specific mode of action of mitochondrial transplantation was not elucidated. The mechanism might be complex considering the various processes, in which mitochondria are involved (e.g., energy production, redox status, apoptosis, calcium modulation), and one specific mechanism might not be responsible for the whole beneficial effects. Despite these limitations, this study is the first to investigate the effects of mitochondria transplantation on sepsis depending on the cell types from which they are isolated.

## 4. Materials and Methods

### 4.1. Mitochondria Isolation from Different Cell Types and Mitochondrial Transplantation

Mitochondrial isolation methods have been previously described [6]. We purified mitochondria from L6 (ATCC; CRL-1458, Manassas, VA, USA), clone 9 (ATCC; CRL-1439, Manassas, VA, USA), and umbilical cord mesenchymal stem cells (UC-MSCs; IRB No. 201806-BR-029-03) by differential centrifugation, which yielded mitochondrial extract as determined by bicinchoninic acid (BCA) assay.

L6 and clone 9 cells were homogenized in SHE buffer (0.25 M sucrose, 20 mM HEPES, 2 mM EGTA, 10 mM KCl, 1.5 mM MgCl2, 0.1% defatted bovine serum albumin [BSA], protease inhibitor, pH 7.4) and then centrifuged at 1500× *g* for 5 min to remove cells and cell debris. After centrifugation, the mitochondria-containing supernatant was subsequently centrifuged at 20,000× *g* for 10 min. UC-MSCs were homogenized in SHE buffers and centrifuged at 1100× *g* for 3 min. The supernatant was centrifuged at 12,000× *g* for 15 min to obtain a mitochondrial pellet and then resuspended the pellet in SHE buffers without BSA. Finally, the mitochondrial suspension was centrifuged at 20,000× *g* for 10 min. All centrifugation steps are performed at 4 °C.

The mitochondrial suspension (in 10 µg/10 μL of PBS) and LPS were added slowly to each tube of recipient cells (1 × 10^5^) suspended in 500 μL of RPMI 1640 supplemented with 10% fetal bovine serum [FBS], 1% penicillin-streptomycin [P/S]. After cell seeding into 48-well plates, we centrifuged it at 1500× *g* for 5 min and then incubated at 37 °C and 5% CO_2_.

### 4.2. ATP Assay

ATP content and ATP synthesis in isolated mitochondria from L6, clone 9, and MSC were assessed using a CellTiter-Glo Luminescent Cell Viability Assay (Promega, Madison, WI, USA). To detect ATP content, 10 μg of isolated mitochondria was incubated with CellTiter-Glo reagent at room temperature for 10 min. To measure ATP synthesis, 5 mM of ADP solution was added to 10 μg of isolated mitochondria and then incubated at 37 °C for 45 min. After incubation, the samples were incubated with CellTiter-Glo reagent at room temperature for 10 min. All experiments were performed protected from light. The luminoscence was measured on a luminometer (BioTex, Winooski, VT, USA).

### 4.3. Stimulation of THP-1 with LPS

The hyperinflammation model was induced by treating THP-1 cells (ATCC, Manassas, VA, USA) with 50 ng/mL lipopolysaccharide (LPS) for 4 h. To estimate the anti-inflammatory effect of mitochondrial transplantation, THP-1 cells were co-incubated with LPS and isolated mitochondria in each cell type. The endotoxin tolerance model was stimulated with 10 ng/mL LPS for 4 h as the first stimulus, followed by resting in a fresh medium without LPS for 16 h. Cells were re-stimulated with 10 ng/mL LPS for another 4 h. To assess the immune-modulation effect of mitochondrial transplantation, cells were treated with purified mitochondria during each LPS exposure. After incubation, cell culture supernatants were collected, and TNF-alpha secretion was measured using enzyme-linked immunosorbent assay (ELISA).

### 4.4. Measurement of Oxygen Consumption Rate

After being cultured for 24 h with LPS and isolated mitochondria, OCR was measured in THP-1 cells using a Seahorse XF analyzer and a Seahorse XF Cell Mito Stress Test kit (Agilent, Santa Clara, CA, USA). THP-1 cells were suspended in Seahorse XF RPMI medium (10 mM glucose, 1 mM sodium pyruvate, and 2 mM L-glutamine) and seeded in Cell-Tak coated XFe96 microplates (1 × 105 cells/well). Cells were then equilibrated in a non-CO_2_ incubator for 1 h at 37 °C. The mitochondrial inhibitors (1.5 µM oligomycin, 2 µM FCCP, and 0.5 µM rotenone and antimycin) were serially injected to each well during the assay. It begins with measuring the base level of OCR, as the complex III inhibitor oligomycin injects, the OCR is rapidly decreased. This will be reversed by the injection of FCCP, an uncoupling agent that can dissipate the proton gradient and maximize the OCR. Finally, followed by the injection of Rotenone/antimycin A, the OCR decreases again. Parameters calculated in the form of a bar chart include ATP-linked respiration, proton leak, basal respiration, maximal respiration, and spare respiratory capacity. Basal respiration shows the energetic demand of cells under basal conditions, the oxygen consumption of basal respiration is used to meet ATP synthesis and results in mitochondrial proton leak. ATP-linked respiration is reflected by the decrease in OCR following the injection of the ATP synthase inhibitor oligomycin, which is the portion of basal respiration. The remaining basal respiration not coupled to ATP synthesis after oligomycin injection represents the proton leak, which can be a sign of mitochondrial damage. Maximal respiration represents the maximum capacity that the electron respiratory chain can achieve. The maximal oxygen consumption rate is measured by the injection of the uncoupler FCCP. Spare respiration is the difference between maximal and basal respiration, which reflects the capability of the cells to respond to changes in energetic demand and indicates the fitness of the cells. Non-mitochondrial respiration is oxygen consumption due to cellular enzymes other than mitochondria after the injection of rotenone and antimycin A [17]. The OCR data were analyzed using the Seahorse Wave 2.6.1 Software.

### 4.5. In Vivo Rat Sepsis Model

This study was approved by the Institutional Animal Care and Use Committee of the authors’ institute (IACUC-220052), in accordance with the National Institutes of Health Guidelines. This study was carried out in compliance with the ARRIVE guidelines. Male Sprague-Dawley rats weighing 270–330 g were used. The rats were housed in a controlled environment (Room temperature 20~24 °C, humidity 40~60%) with access to standard food and water ad libitum for 7 days before the experiment.

We used a body weight-adjusted polymicrobial sepsis model according to a previous study [18]. In brief, we used inhalation anesthesia with isoflurane for the short term and then injected intramuscular Zoletil (50 mg/kg) and Xylazine (10 mg/kg) before experiments. Feces were collected from donor rats. A midline laparotomy was performed, and the cecum was extruded. A 0.5 cm incision was made in the antimesenteric surface of the cecum and it was squeezed to expel feces. The collected feces were weighed and diluted with 5% dextrose saline at a ratio of 1:3. This fecal slurry was vortexed to make a homogeneous suspension before administration into the intraperitoneal cavity. In sepsis induction, rats were anesthetized as above, and 0.5 cm midline laparotomy was performed, and fecal slurry was administered into the peritoneal cavity. The volume of fecal slurry given to each animal was adjusted on the body weight of the recipient rat. We administered subcutaneous fluid resuscitation (30 mL/kg 5% dextrose saline) and imipenem was injected subcutaneously at a dose of 25 mg/kg twice daily for 2 days. We did not use painkillers. Thereafter, the rats were randomly assigned to 4 groups, mitochondrial transplantation (L6, clone 9, MSC) and control groups (Figure 2A). Randomization was done by a research assistant, who was not performing the main procedure. The researcher who was performing the main procedures was blind to the allocated groups. The body weight of rats could be a confounder, so we randomized it with the body weight-stratified method. We did not think other confounders could affect the results with this study design, so it was not controlled.

Mitochondria or DPBS were administered 1 h after sepsis induction at a dose of 200 µg via the tail vein. Survival was monitored every 12 h for 14 days. During the observation period, an employee of the animal research center monitored animals twice per day, and if the animal seemed close to death, they notified the research team, who made a decision for euthanasia.

### 4.6. Quantification of Cytokines

The levels of the cytokines TNF-α (ab181421, Abcam, Waltham, MA, USA) in THP-1 cell homogenates were measured using ELISA kits according to the manufacturer’s instructions. The optical density at 450 nm was detected by a VersaMax microplate reader (SoftMax Pro 7.1 software, Molecular Devices, San Jose, CA, USA).

### 4.7. Serum Alanine Aminotransferase (ALT) and Creatinine Concentration in Plasma

The ALT and creatinine levels in the plasma sample were determined by an automated chemistry analyzer (Beckman Coulter, Brea, CA, USA) [19].

### 4.8. Arterial Blood Gas Analysis

Arterial blood samples were analyzed using an ABL 90 blood gas analyzer (radiometer, Copenhagen, Denmark). Blood was drawn into a syringe containing heparin and analyzed within 10 min or cooled immediately [19].

### 4.9. Statistical Analysis

Data were expressed as mean ± standard deviation (SD). The normality for the distribution of variables was verified by the Shapiro–Wilk test. A One-way ANOVA with Bonferroni post hoc tests was used to compare means in the normally distributed data. If the distributions are not normal, the data were analyzed using the Kruskal–Wallis test. Survival probabilities were calculated using the Kaplan–Meier method. All *p*-value < 0.05 was considered significant. Statistical analysis was performed using EZR software, version 1.54 (Saitama Medical Center, Jichi Medical University, Saitama, Japan).

## 5. Conclusions

The effects of mitochondria transplantation on both in vitro and in vivo sepsis models differed based on the cell types of origin of the mitochondria. L6-mitochondria showed improved mitochondrial function when transplanted in an in vitro sepsis model in terms of oxidative phosphorylation. Furthermore, it demonstrated a greater survival effect in the in vivo sepsis model, compared to MSC or clone 9 cells.

## 6. Patents

KK, TNC, YSC, and MJL are inventors on a patent application related to this study (Application No. 10-2019-0050017 (Republic of Korea) Filed on 29 April 2019).

## Figures and Tables

**Figure 1 ijms-24-10113-f001:**
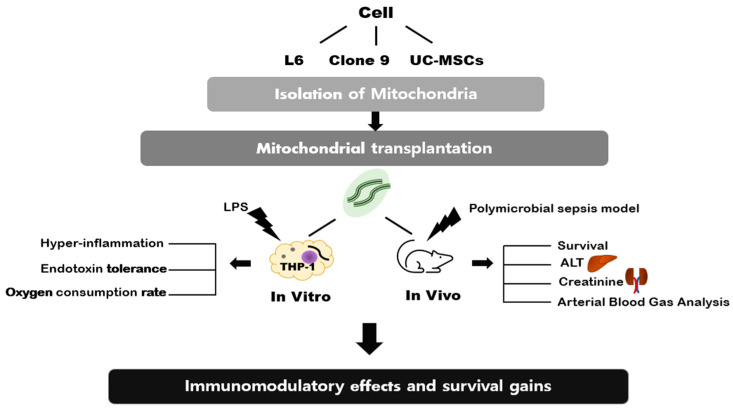
Scheme of in vitro and in vivo models for studying mitochondrial transplantation.

**Figure 2 ijms-24-10113-f002:**
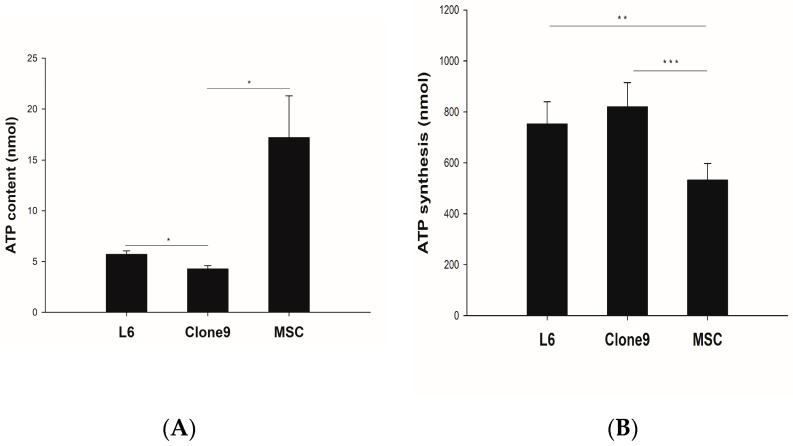
Characteristics of isolated mitochondria from different cell lines. To compare the function of the isolated mitochondria from L6, clone 9, UC-MSC cells, (**A**) ATP content and (**B**) ATP synthesis of isolated mitochondria were measured using luminescence-based assay. The data were presented as mean ± SD. Statistical analysis was performed using Kruskal–Wallis and One-way ANOVA and * *p* < 0.05, ** *p* < 0.01, *** *p* < 0.001.

**Figure 3 ijms-24-10113-f003:**
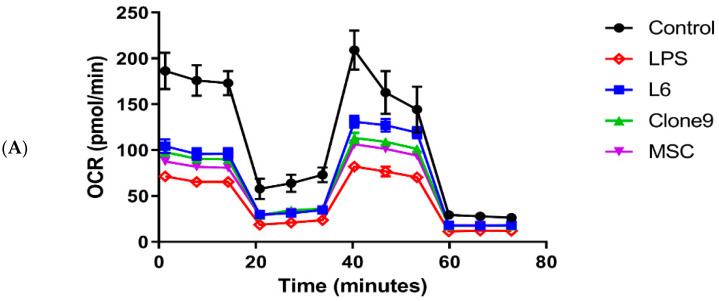
Oxygen consumption rate (OCR) was measured in the hyper-inflammatory in vitro model after mitochondrial transplantation. (**A**) OCR levels were measured using a Seahorse XF analyzer with oligomycin (1.5 µM), FCCP (2 µM), rotenone, and antimycin (0.5 µM) in LPS induced hyper-inflammatory THP-1 cells following mitochondrial transplantation with three different cell types (L6, Clone9, MSC). (**B**) The rate of ATP production, basal respiration, maximal respiration, non-mitochondrial oxygen consumption, proton leak, and spare respiratory capacity. Control means THP-1 cell without LPS stimulation. Results are presented as the mean ± SD and significant differences between groups, * *p* < 0.05, ** *p* < 0.01, *** *p* < 0.001.

**Figure 4 ijms-24-10113-f004:**
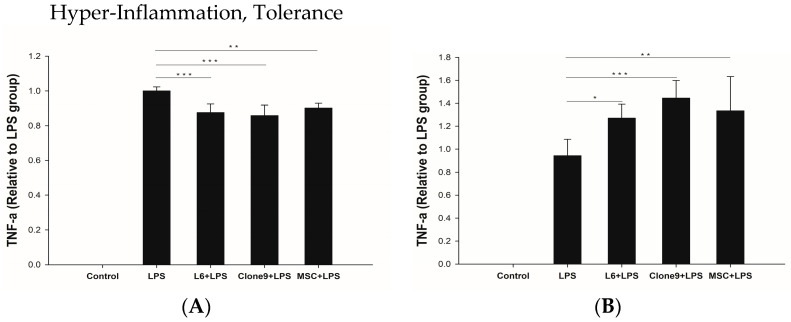
Immunomodulatory effect of mitochondrial transplantation in LPS-stimulated THP-1 monocytes was examined. (**A**) In hyperinflammation model, mitochondrial transplantation significantly decreased the TNF-a level. (**B**) Conversely, in the endotoxin tolerance model, TNF-a level was increased following mitochondrial transplantation. Control means THP-1 cell without LPS stimulation. Results are presented as the mean ± SD and significant differences compared with LPS group, * *p* < 0.05, ** *p* < 0.01, *** *p* < 0.001.

**Figure 5 ijms-24-10113-f005:**
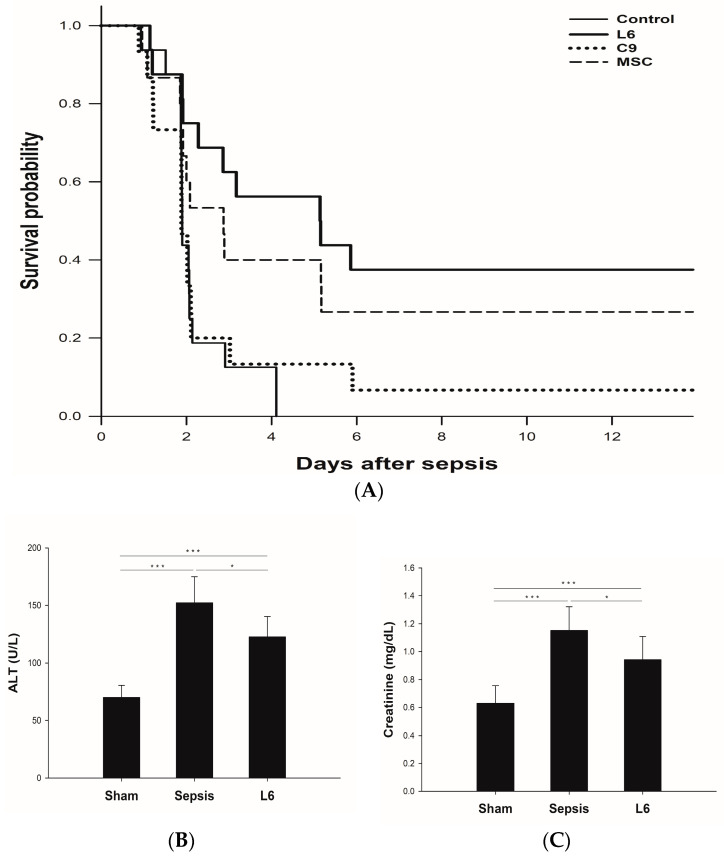
The therapeutic effect of mitochondrial transplantation was assessed using an in vivo fecal slurry rat model. (**A**) Kaplan–Meier survival curves revealed that only L6 mitochondrial transplantation significantly increased survival compared to control. (**B**) Plasma alanine transaminase (ALT) and (**C**) creatinine levels were measured after 24 h of fecal slurry model. Both ALT and serum creatinine were reduced following mitochondrial transplantation. The results shown were the mean ± SD. Statistical analysis was performed using One-way ANOVA and * *p* < 0.05, *** *p* < 0.001.

## Data Availability

The datasets generated and analyzed during the current study are available from the corresponding author on reasonable request.

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
