# Peer review of "The Effects of Mitochondrial Transplantation on Sepsis Depend on the Type of Cell from Which They Are Isolated"

_ijms, 2023, doi:10.3390/ijms241210113_

Round 1

Reviewer 1 Report

An interesting paper with clear clinical/translational relevance. The manuscript should be improved as follows. Considering that many readers will not be familiar with the concept, a brief definition of mito transplantation should be given in the Introduction. The recent review by Mokhtari PMID 35706715 should be considered in the Introduction. Axis labeling of the figures needs to be improved, e.g., larger font and in the case of x-axes, it needs to be clearer whether for instance "L6" next to "LPS" indicates that L6 mitochondria were applied to LPS-stimulated cells (e.g., LPS+L6). In the cellular model, why was only TNF assayed? At least one other cytokine, e.g. IL-6, should be measured. In 2.5. the description/interpetration of results is insufficient, in 2.6 it is missing altogether. In Methods, provide more detail how the mitochondria were applied to cells (transfected? just added to cell culture media?) and infused into the animal (e.g., at what rate, etc.).

Still some grammatical and typographical errors to be corrected.

Author Response

Dear Reviewer

I really appreciate your great comments. It definitely improved my research and manuscript.

I attached detailed response as word file.

Sincerely your

Dr Kim

Reviewer 2 Report

The work of Yun-seok Kim and his collaborators is very interesting because it addresses the very important issues of sepsis and the effect of mitochondrial origin and its ability to protect the organism against sepsis. The study is conceptually very well laid out, but there are some very important issues that need to be addressed before it can be published.

The English language is very poor and the paper needs to be written in academic English. Sometimes it is very difficult to understand what the authors wanted to say. I suggest taking the help of a native speaker or an authorized translator.

The title: it is "The Effects of Mitochondrial Transplantation on sepsis in dependent on the origin cell of mitochondria isolation". Perhaps the authors should consider changing the title. Suggestion: "The effects of mitochondrial transplantation on sepsis depend on the type of cell from which they are isolated"

Abstract:
- The origin of some cell types is explained, but not for the THP-1 cell line. This should be consistent.
- Lane 24: It says: "First, we saw the mitochondrial function in mitochondrial transplantation." What did the authors mean by this? The sentence is not clear and precise enough.

The Introduction section:
- The introduction lacks an explanation of why the selected cells were used. Although the origin of the cells is given in the abstract, it would be desirable to give the exact and complete names of all cells used and their abbreviations. In the Discussion section (lines 171-179), there is a paragraph about the cells, but this should be replaced in the Introduction

- Lanes 58-61: I suggest rewriting the text as follows: With this in mind, we hypothesized that the effects of transplanting mitochondria would vary depending on the cell type from which they were isolated. To test this hypothesis, we used both in vivo and in vitro sepsis models. We examined the immunomodulatory effects and survival of mitochondrial transplantation using different cell types for mitochondrial isolation (Fig. 1).

  The results:
- Lanes 74-78: The legend under Figure 2 contains spelling and grammatical errors.
- Lane 77: Fonts in the text should be consistent.
- A heading for the X-axis is unnecessary. Please delete the word "group" under the X-axis in all diagrams.
- Lane 105: It says: "The rate of ATP production, basal respiration, maximal respiration, non-mitochondrial oxygen consumption, proton leak and spare respiratory capacity." - these methods are not explained in the materials
- Lane 124: In Figure 4, under A and B, what is control and why is it set to 0? - Lane 135: It is said "We confirmed the effect of isolated mitochondria in survival experiments of a polymicrobial sepsis model using Rat", please rephrase this sentence, what does this "RAT" mean? It is not precise enough.
- Lane 146: Figure 5 - where are the ALT and CREATININE graphs for other cell lines when the text says it is measured in all cell lines
- Lane 154: It says: "Plasma creatinine, ALT, and ABGA were measured obtained from the L6-treated sepsis model. All values favor the mitochondrial transplantation in sepsis model." - Why did the authors choose only this model if they wanted to compare all three cell lines?
- Lane 153 - Section 2.5. Plasma Chemistry Assay: ALT Albumin, Blood Urea Nitrogen (BUN), Creatinine, and Lactate. Where are the results for these analyzes?

Discussion
- In this section, the authors refer to oxidative phosphorilation, but there is no information on oxidative enzymes. I suggest providing more results on oxidative phosphorilation to discuss this aspect.
- This paragraph looks like the introduction and should be rewritten to comment on the results.  

The general impression is that the paper is poorly written and can not be considered for publication in the present form.
All paragraphs should be rewritten.

The English language is very poor and the paper needs to be written in academic English. Sometimes it is very difficult to understand what the authors wanted to say. I suggest taking the help of a native speaker or an authorized translator.

Author Response

(The authors gave the same response as above.)

Round 2

Reviewer 1 Report

There are appears to be a reference to a review on mitochondrial transplantation in Introduction now, but no reference is cited.

It is still not clear whether 2.5 refers to any results. Could it be the preceding figure? 

Still a fair number or English errors. But the overall message is understood. 

Author Response

We appreciate your detailed and helpful comments.

We corrected as follows;

There are appears to be a reference to a review on mitochondrial transplantation in Introduction now, but no reference is cited.

-->We apologize for mistake. We inserted reference number.

It is still not clear whether 2.5 refers to any results. Could it be the preceding figure? 

-->As you recommended, we insert the fig 5-B, C to the 2.5 text.

Reviewer 2 Report

Dear authors,
Thank you for accepting the suggestions and significantly improving the quality of the work.
The method of mitochondrial transplantation should have been better explained. Nevertheless, the paper is very interesting and has been improved.

English has been improved.

Author Response

We appreciate your helpful comment.

As you recommended, we correct as follows;

The method of mitochondrial transplantation should have been better explained. 

-->We added more detailed method.